# Unveiling the digital revolution: Catalyzing total factor productivity in agriculture

Jing He[1], Zhuangyu Wei[2]*, Xiaokai Lei[3]

**1** College of Agriculture, Guangxi University, Nanning, Guangxi, China, **2** School of Economics, Guangxi University, Nanning, Guangxi, China, **3** Brandeis International Business School, Brandeis University, Boston, Massachusetts, United States of America

\* gxnnwzy123@163.com

## Abstract

Drawing upon panel data spanning the years 2011 to 2022 and encompassing 30 provinces across China, this research employs empirical methodologies, specifically the difference GMM and system GMM methods, to scrutinize the impact of the digital economy on the total factor productivity (TFP) within the agricultural sector. The study reveals a significant augmentation of China's agricultural TFP attributable to the digital economy, a finding robust to various methodological examinations. Notably, the influential role of the digital economy on agricultural TFP is more conspicuous in the central and western regions, as well as in locales characterized by lower productivity levels. Mechanistic analysis underscores that the digital economy exerts a positive influence on agricultural TFP through the stimulation of innovation and marketization effects. Furthermore, strategic recommendations emerge from this study, advocating for the reinforcement of institutional and mechanistic reforms to cultivate an enabling external milieu for the digital economy to propel agricultural TFP. It is posited that regional development strategies should be tailored based on individual resource endowments and the extent of digital economic development. Additionally, there is a call to refine mechanisms promoting high-quality development in agriculture, with an overarching goal of comprehensively elevating agricultural TFP. The implications of this research extend to the imperative need for a nuanced and context-specific approach to advancing agricultural productivity across diverse regions in China.

## Introduction

Since the initiation of the reform and opening-up policy, China's agricultural sector has undergone rapid development, culminating in a noteworthy total grain output surpassing 695 million tons by the year 2023. This achievement is particularly significant as it has maintained a consistent level exceeding 650 million tons for night consecutive years, attaining an unprecedented pinnacle in historical records. However, concomitant with the progress in China's economic development and the enhancement of people's quality of life, a discernible transformation has transpired in the consumer demand structure for agricultural products [1].

This evolution is characterized by an escalating demand for high-quality agricultural products, juxtaposed against a surplus of low-quality counterparts [2]. Consequently, a pronounced and progressively exacerbating contradiction between supply and demand has

**Data availability statement:** In our manuscript, "Unveiling the digital revolution:

Catalyzing total factor productivity in agriculture", all relevant data are within the paper and its Supporting Information files. The data for the dependent variable, core explanatory variables, and control variables are sourced from the China Economic Net Statistical Database, the "Compilation of Statistical Data for 50 Years of China," annual editions of the "China Statistical Yearbook," the "China Industrial Economic Statistical Yearbook," the "China Provincial Marketization Index Report," the "China Regional Innovation Capacity Assessment Report," and the statistical yearbooks of individual provinces. The authors confirm that all data underlying the research findings described in the manuscript are fully available and unrestricted.

**Funding:** The author(s) received no specific funding for this work.

**Competing interests:** The authors have declared that no competing interests exist.

emerged at both ends of the agricultural product spectrum [3]. This disjunction underscores the imperative to address the evolving dynamics of consumer preferences and the ensuing challenges faced by the agricultural sector. A nuanced understanding of these shifting patterns is crucial for formulating strategic interventions that can effectively align agricultural production with the contemporary demands of an evolving consumer landscape.

In this context, General Secretary Xi Jinping has underscored the imperative of augmenting the comprehensive production capacity of agriculture and advancing the high-quality development of social security [4]. Recognizing agriculture as a foundational pillar of the national economy, it assumes a pivotal role in buttressing the construction and advancement of the national economy, holding critical significance for ensuring the well-being of the populace and attaining societal stability [5]. Against the backdrop of China's evolving digital landscape, the State Council, in 2022, promulgated the "14th Five-Year Plan for Advancing Agricultural and Rural Modernization." This strategic framework accentuates the expeditious digitization of agriculture and rural areas, the pursuit of key core technologies in agriculture and agricultural machinery, the profound restructuring of the supply side of agriculture, and the facilitation of a comprehensive upgrade within the agricultural sector.

The ascendancy of the digital economy not only injects fresh momentum into China's economic trajectory but also serves as a transformative force propelling a new phase of evolution across diverse industries. As the digital paradigm permeates the agricultural domain, its ramifications extend beyond economic dimensions to encompass societal well-being and stability. The strategic emphasis on digitization aligns with a broader vision that acknowledges the catalytic role of technological advancements in steering a substantive and multifaceted transformation across the agricultural landscape and, by extension, the broader national socio-economic fabric.

Thus, a pivotal inquiry arises: can digital economy effectively increase/boost the TFP of Chinese agriculture industry? A precise and well-founded response to this question holds the potential to furnish a theoretical underpinning for China's future endeavors in advancing digital economic development and fortifying its stature as an agricultural powerhouse. The exploration of this nexus between the digital economy and agricultural productivity stands not only as an academic pursuit but also as a strategic imperative, guiding the formulation of policies and initiatives that resonate with the broader objectives of economic progress and agricultural resilience.

## Literature review

Presently, scholarly attention is directed towards the meticulous measurement and analysis of the digital economy. Crawford undertakes a comprehensive exploration of the measurement methods applied to free digital services [6]. Meanwhile, Xu and Zhang, through rigorous calculations, observe a nuanced trajectory in the proportion of China's digital economy value-added to GDP, noting an initial decline followed by a subsequent ascent spanning the years 2007 to 2017, with a pronounced upward surge post-2012 [7]. Jiang et al., in their examination of the present status and characteristics of the digital economy, posit its pivotal role in augmenting entrepreneurial activities, propelling consumption growth, and optimizing resource allocation in the Chinese context [8]. Another cohort of scholars delves into the broader impacts of the digital economy on social and economic domains. Wang and Shen scrutinize the interplay between the development of the digital economy and China's labor structure, revealing that digital economy development heightens the demand for labor holding a bachelor degree or higher while concurrently diminishing the demand for labor with a high school education or below [9]. Huang et al. ascertain that the digital economy serves as a catalyst for high-quality development in China's economy, leveraging mechanisms leveraging

mechanisms in areas such as factor inputs, resource allocation efficiency, and production efficiency [10]. Building on this, Wu et al. uncover that the digital economy instigates transformative shifts in traditional corporate governance paradigms. It propels high-quality development in China's real economy by expanding the boundaries of division of labor, curbing transaction costs, amplifying the space for product value-added, and markedly elevating the levels of green innovation within enterprises [11].

Numerous scholars have devoted their efforts to the examination of China's agricultural TF, yielding a wealth of academic insights. Gao conducted a thorough analysis of the connotation, current developmental status, and challenges confronting agricultural TFP [12]. He underscored that technological progress, harnessing competitive advantages, and prioritizing the construction of industrial chains constitute positive drivers for the enhancement of agricultural TFP. In a complementary endeavor, Liu et al. devised an indicator system for agricultural TFP grounded in the new development concept. Their scrutiny of China's agricultural TFP from 2003 to 2016 revealed existing weaknesses in agricultural development, such as low levels of coordination and openness. Simultaneously, they identified green development, innovation, and pivotal technologies as instrumental elements in elevating China's agricultural TFP [13]. Li and Li contributed to the discourse by calculating the agricultural TFP across 31 provinces in China spanning the years 2002 to 2016. Their findings underscore a steady increase in provincial agricultural TFP, accompanied by a gradual narrowing of disparities between provinces [14]. Exploring the urban-rural interface, Gong et al. revealed a significant correlation between agricultural TFP and the urbanization process. However, they observed a weak inhibitory effect of the new urbanization paradigm on the growth efficiency of agricultural TFP. Collectively, these scholarly endeavors enrich our understanding of the multifaceted dynamics shaping China's agricultural TFP landscape [15].

A cadre of scholars has undertaken research endeavors to elucidate the impact of the digital economy on agricultural TFP. The digital economy emerges as a transformative force that not only mitigates the information asymmetry between rural and urban areas but also fosters the expansion of agricultural production scale, bolsters resource utilization efficiency, and augments farmers' income. In effect, this catalyzes the modernization on both agriculture and rural areas in the country [16]. Wang et al. delved into the influence of digital economic development on production efficiency, revealing a positive impact between 2011 and 2020. However, in comparison to the service and industrial sectors, the promoting effect of the digital economy is notably less pronounced in agriculture [17]. Yang et al. emphasized the constructive role played by the digital economy in the digitization of agriculture, pinpointing its precise targeting of agricultural inputs, facilitation of intelligent agricultural decision-making, transparency in transaction processes, and dissemination of agricultural knowledge [18]. In an empirical study, Xue et al. investigated the impact of digital technology on China's agricultural TFP, establishing that digital technology exerts a significantly positive influence on the agricultural TFP across various provinces in China, with the most pronounced effects observed in the eastern and central regions [19]. Examining China's agricultural landscape in a global context, Xia et al. drew comparisons with agriculturally developed countries such as the United States, France, Germany, and Japan. Their analysis delineated a discernible gap in China's agricultural sector vis-à-vis these countries. Agriculturally advanced nations have leveraged the digital economy to optimize agricultural industry systems, revamp production paradigms, and enhance management systems. This comprehensive approach not only amplifies resource utilization efficiency and market competitiveness but also fosters heightened productivity and management efficiency [20]. Collectively, these scholarly investigations contribute to a nuanced understanding of the intricate interplay between the digital economy and agricultural TFP.

The extant literature has undertaken a comprehensive exploration of the measurement and developmental status of the digital economy, scrutinizing its effects and delving into the construction of indicators for agricultural TFP. Additionally, scholars have investigated influencing factors from diverse perspectives, collectively laying a robust theoretical foundation for understanding the parallel growth trajectories of the digital economy and agricultural TFP. Nevertheless, there are certain gaps that persist in the existing body of knowledge. Firstly, the existing research lacks a precise delineation of the role played by the digital economy in promoting agricultural TFP. Secondly, the majority of studies hitherto have approached the measurement and analysis of China's agricultural TFP primarily from a theoretical standpoint, creating a conspicuous void in empirical investigations that directly probe into the intricate relationship between the digital economy and agricultural productivity. Addressing these gaps is imperative for refining our comprehension of the dynamic interplay between the digital economy and the multifaceted dimensions of agricultural TFP.

Recognizing the identified gaps in the existing literature, this paper undertakes a comprehensive review of the theoretical mechanisms underlying the influence of the digital economy on agricultural TFP. To empirically investigate these dynamics, panel data spanning 30 provinces in China from 2011 to 2022 is employed. Through rigorous empirical testing, the research seeks to delineate the impact of the digital economy on agricultural TFP. The overarching objective is to contribute to the augmentation of scholarly understanding regarding the nuanced ways in which the digital economy propels agricultural TFP in the context of China. By bridging these empirical gaps, this study aims to furnish valuable insights that inform and refine our comprehension of the intricate relationship between the digital economy and the multifaceted dimensions of agricultural productivity within the Chinese landscape.

## Hypothesis development

### The innovative incentive effects of the digital economy on agricultural TFP

Derived from the foregoing literature review, it is discernible that the digital economy has evolved into a pivotal instrument for mitigating diverse contradictions and adverse factors within China's agricultural sector. With the ongoing deepening of integration between the digital economy and agriculture, the innovative incentive effects on TFP can be succinctly summarized as follows:

Primarily, the evolution of the digital economy serves as a catalyst for technological innovation in agriculture, a key driver in enhancing TFP within the sector. This transformative process shifts the conventional labor-intensive agricultural development paradigm towards a novel model propelled by technological innovation [21]. As China's agricultural development encounters bottlenecks, The simple infusion of factors alone is insufficient to meet the growing demand for high-quality agricultural products. The suboptimal quality of agricultural products and inefficient innovation emerge as pivotal obstacles impeding the overall advancement of agricultural productivity in China [22]. In this context, the digital economy assumes a dual role, augmenting both the scale of agricultural production and management efficiency, and concurrently broadening the financing channels for innovation. This contributes to an improved innovation atmosphere and reduced innovation risks in agriculture [5,6]. China, currently positioned in the initial phase of transitioning from an agricultural giant to an agricultural powerhouse, benefits from the digitalized research and development management system. This system contributes to the establishment of an open innovation framework spanning the entire agricultural industry chain, harnessing the spillover effects of digital technology innovation [23]. Technologies such as mobile internet, artificial intelligence, and big data play pivotal roles in allowing innovation entities to precisely locate resources for innovation.

This facilitates research and development activities across diverse scenarios, thereby enhancing the precision of innovation in agricultural production modes [24]. This synergy between the digital economy and technological innovation not only addresses the limitations hindering traditional agricultural models but also positions China on a trajectory towards pioneering advancements in agricultural productivity.

Secondly, the evolution of the digital economy instigates a transformative shift in the production methods and innovation models of agricultural enterprises. In the traditional production processes of agricultural enterprises, the dichotomy between production tools and production data prevailed. However, the advent of digital technologies such as the Internet of Things, cloud computing, and big data has revolutionized the production methods of agricultural enterprises comprehensively [25]. With the gradual permeation of the digital economy across various production stages, there is a discernible transformation of tangible material production factors into intangible ones. This metamorphosis allows production factors embedded in digital information to achieve efficient mobility, facilitating the streamlining of production processes, reduction of operational costs, and enhancement of resource utilization efficiency [26]. Furthermore, traditional enterprise innovation models were predominantly centered around the enterprise itself, often detached from direct consumer interaction. However, within the ambit of the digital economy, the emergence of new production entities such as the Internet, social platforms, and e-commerce narrows the gap between enterprises and consumers. This convergence not only strengthens the connection between supply and demand but also enhances the alignment between businesses and consumers. Consequently, these digital channels become pivotal sources of innovation for enterprises, stimulating diverse and personalized consumer demands. This dynamic interaction enables enterprises to swiftly discern customer needs and feedback, facilitating timely product and technology upgrades. Thus, these digital channels assume a critical role in propelling innovation within enterprises [27]. The confluence of the digital economy and agricultural enterprise dynamics, not only revolutionizes production methodologies but also invigorates innovation models, fostering a more responsive and consumer-oriented paradigm.

## The digital economy has market-oriented effects on the overall factor productivity in agriculture

In addition to its direct impact on agriculture, the maturation of the digital economy also exerts influence on overall factor productivity in agriculture through external market-oriented effects.

The progression of the digital economy, firstly, heightens market competition, prompting inefficient agricultural enterprises to optimize resource allocation and thus elevate overall factor productivity in agriculture. The pervasive influence of the digital economy across diverse industries serves to diminish information asymmetry and substantially augment the fluidity of factors [28]. Within the market's natural selection mechanism, efficient agricultural enterprises find themselves compelled to perpetually augment investments in high-quality production factors to solidify their positions, concurrently augmenting overall production efficiency [29]. The market, endowed with the ability to discern the quality of enterprise products, exerts a constraining influence on low-quality goods, compelling inefficient enterprises to exit the market. Consequently, traditional mid- to low-end agricultural enterprises, characterized by low value-added and inefficient production activities, encounter a further contraction of their limited survival space. External environmental pressures impel these traditional mid- to low-end enterprises to undergo transformation and upgrade, thereby enhancing factor allocation efficiency [25]. This process not only streamlines the market landscape but also propels a

structural shift within the agricultural sector, compelling enterprises to evolve towards greater efficiency and competitiveness.

Secondly, the evolution of the digital economy has induced transformative changes in the market supply-demand structure, consequently imparting a positive influence on overall factor productivity in agriculture. On one hand, the development of the digital economy serves as a catalyst for the diversification of consumer demands. As the digital economy advances, the erstwhile unidirectional flow of products from suppliers undergoes a paradigm shift towards a bidirectional communication between supply and demand [30]. This evolution fosters increased social interactivity between businesses and consumers, stimulating a demand for product diversification. This surge in demand for diverse products, coupled with an expansion in product variety, acts as a driving force for enhancements in both product quantity and quality. On the other hand, the digital economy optimizes transactional models for products. The proliferation of product quantity and variety generates a substantial volume of information, introducing complexity into economic and social systems and giving rise to new transactional challenges [31]. Emerging digital technologies such as mobile internet, big data, and cloud computing, among others, play a pivotal role in addressing these transactional challenges within the economic market. By reducing search costs and enhancing information transmission capabilities, these technologies effectively navigate the complexities introduced by the redundancy of information [32]. For instance, in e-commerce platforms, advertisers leverage big data analytics technology to deliver highly relevant ads based on consumers' click rates and search preferences. This targeted approach significantly diminishes the likelihood of consumers encountering irrelevant ads, thereby enhancing transaction efficiency.

Based on the aforementioned analysis, this paper posits the following hypotheses.

Hypotheses: All other conditions being equal, the level of development in the digital economy is significantly positively correlated with overall factor productivity in agriculture.

## Research design

### Measurement modelling

To account for potential inertia and local adjustments in *TFP* growth, we incorporate the first-order lag of *TFP* (L.*TFP*) into the model. This inclusion enables us to construct a dynamic panel model to evaluate the digital economy's impact on agricultural TFP.

$$TFP_{it} = \lambda TFP_{it-1} + \alpha Digital_{it} + \sum_{j} \beta_j X_{ijt} + u_i + \gamma_t + \varepsilon_{it} \tag{1}$$

In equation (1), $TFP_{it}$ denotes the level of overall factor productivity in agriculture; $Digital_{it}$ represents the degree of digital economy development; $X_{ijt}$ represents control variables; the subscript *i* denotes the province, *j* denotes the control variable index, and *t* denotes the year. The terms $u_i$ and $\gamma_t$ capture province and time effects, respectively; $\varepsilon_{it}$ represents the error term.

### Variable selection

**Agriculture TFP.** To calculate agricultural TFP, we follow the approach of Lin and Gong, for selecting input and output indicators [33,34]. The output indicator is the gross output value of agriculture, forestry, animal husbandry, and fisheries (in ten thousand RMB), adjusted to constant 2011 prices using the agricultural GDP deflator. Input indicators include total machinery power (ten thousand kilowatts), fertilizer consumption (ten thousand tons), rural labor employment (ten thousand persons), and total crop sown area (thousand hectares).

Methodologically, because DEA and SFA methods can only capture the contemporaneous growth rate of TFP rather than its level, we focus on analyzing the digital economy's impact on agricultural TFP (TFP) levels. To achieve this, we use the classic Solow residual method and the fixed effects method, denoted as TFP_OLS and TFP_FE, respectively. To account for time-related influences, we also control for time-fixed effects in the estimation. In the empirical analysis that follows, we report the estimation results for all regressions with TFP_OLS and TFP_FE as dependent variables.

**The level of development in the digital economy.** This study follows Dian et al. by constructing a comprehensive index system to assess the development of China's digital economy, focusing on two dimensions: digital industrialization and industrial digitalization [35]. The specific indicators are listed in Table 1. For digital industrialization, we use indicators including internet penetration, revenue from the manufacturing of computers, telecommunications, and other electronic devices, software industry revenue, per capita telecom usage, and mobile phone users per hundred people. For industrial digitalization, we use indicators including China's Digital Inclusive Finance Index, the share of enterprises engaged in e-commerce, e-commerce procurement value, and e-commerce sales. In the calculations, a combination of principal component analysis and entropy weighting is applied to mitigate the dimensionality of the indicators. The outcomes are denoted as "*Digital*" and "*DigitalA*," earmarked for baseline regression and robustness testing, respectively. We use principal component analysis and the entropy weight method for dimensionality reduction, with the results denoted as *Digital* and *DigitalA*, respectively, applied in the baseline regression and robustness checks.

**Control variables.** To reduce the impact of omitted variables on estimation results and isolate the effect of key factors, we control for various potential influences on agricultural TFP. The theoretical relationships between these factors and agricultural TFP are outlined below.

**Urbanization (URBAN).** The impact of urbanization on agricultural TFP can be seen in two ways. On one hand, it encourages labor migration from agriculture to non-agricultural sectors, driving improvements in production efficiency and resource utilization. Second, urbanization promotes technological diffusion and boosts market demand, thereby incentivizing agricultural innovation. On the other hand, urbanization may cause rural labor outflow and a reduction in arable land, leading to lower agricultural output and TFP. This study measures urbanization using the proportion of urban population to total regional population.

**Industrial upgrading (IU).** Industrial upgrading reallocates resources from the low-efficiency traditional agricultural sector to more productive modern industries. The growth of

Table 1. The indicator system for the level of development in the digital economy.

| The target level | The criteria level | Indicators |
|---|---|---|
| The level of development in the digital economy | Digital industrialization | Internet penetration rate |
| | | Main business revenue of computer, communication, and other electronic equipment manufacturing |
| | | Software industry revenue |
| | | Per capita telecommunication service volume |
| | | Mobile phone users per hundred people |
| | Industrial digitization | China's Inclusive Digital Finance Index |
| | | Proportion of enterprises engaged in e-commerce transactions |
| | | E-commerce procurement amount |
| | | E-commerce sales amount |

the secondary and tertiary sectors also offers increased technological support to agriculture, boosting agricultural TFP. We define industrial upgrading using the following equation (2).

$$UI_{it} = \sum_{i=1}^{3} x_i \cdot i, 1 \leq i \leq 3 \tag{2}$$

The variable $x_i$ denotes the share of industry i's output in total output.

**The degree of dependence on foreign trade (OPEN).** Import and export trade may influence agricultural TFP in both directions. A higher reliance on foreign trade allows China to benefit from technology spillovers, adopting advanced agricultural technologies and management practices from developed countries. Additionally, international market competition can drive efficiency gains and improve product quality in China's agricultural sector. However, greater reliance on foreign trade may expose agricultural prices to international market fluctuations, undermining the stability of the sector. Import and export trade may also redirect resources to non-agricultural sectors, crowding out agricultural inputs. We measure foreign trade dependence as the ratio of total imports and exports to GDP, with trade values converted using the annual average exchange rate between the US dollar and the Chinese yuan.

**The share of irrigated area (IA).** An increase in the share of irrigated area improves water supply to farmland, enhances land use efficiency, and boosts crop yield and quality, thereby positively impacting agricultural TFP. However, excessive expansion of irrigated area may lead to water waste and other environmental issues, ultimately undermining agricultural TFP. We define IA as the ratio of irrigated farmland to total crop area

**Rural human capital (HUM).** Rural human capital impacts agricultural TFP in two key ways. First, high-quality human capital fosters the adoption and innovation of agricultural technologies, accelerates the spread of modern practices, and boosts productivity. Second, improvements in human capital can facilitate the shift from labor-intensive to knowledge-intensive agriculture, increasing the value added to production. Rural human capital is measured by the average years of education for the population aged 6 and older, as specified in equation (3).

$$HUM = \sum_{i=1}^{n} p_i y_i \tag{3}$$

In equation (3), $p_i$ represents the years of education for each level, and n denotes the number of education categories, set to 5. Specifically, we categorize the population's education level into five groups: illiterate or functionally illiterate $y_1 = 0$ primary education $y_2 = 6$ junior secondary education $y_3 = 9$ senior secondary or vocational education $y_4 = 12$ and tertiary education or higher $y_5 = 16$

The share of crop disaster-affected area (DISA). A higher proportion of crop disaster-affected area suggests significant damage to arable land, crops, and agricultural infrastructure. This, in turn, negatively affects agricultural output and resource allocation efficiency. We measure DISA as the ratio of affected crop area to total arable land.

**Government investment (LGOV).** Government investment in agriculture is primarily directed toward infrastructure development, technological research and development, and subsidies for agricultural products. By increasing such investment, production conditions improve, technological progress is accelerated, and agricultural TFP is enhanced. However, it is crucial to note that the effectiveness of government investment is shaped by factors such as policy design, implementation quality, and resource allocation. Over-reliance on direct subsidies or imprecise targeting can result in resource misallocation and incentive distortions

in the agricultural sector, ultimately impeding improvements in agricultural TFP. We measure government investment as the expenditure on agriculture, forestry, and water affairs per unit of sown area. Given the high variability of this indicator, we apply a natural logarithm transformation.

**Mechanism variables. Regional innovation capacity (INNO).** The "China Regional Innovation Capacity Assessment Report" provides ratings across five areas: knowledge creation, knowledge acquisition, corporate innovation, innovation environment, and innovation performance. We use the composite scores from this report to assess the innovation capacity of each province.

**The level of marketization (MKT).** We measure marketization using the index provided in the "China Provincial Marketization Index Report."

## Data sources and processing notes

Our analysis covers 30 provinces from 2011 to 2022. The data for the dependent variable, core explanatory variables, and control variables are sourced from the China Economic Net Statistical Database, the "Compilation of Statistical Data for 50 Years of China," annual editions of the "China Statistical Yearbook," the "China Industrial Economic Statistical Yearbook," the "China Provincial Marketization Index Report," the "China Regional Innovation Capacity Assessment Report," and the statistical yearbooks of individual provinces.

The descriptive statistics for each variable are provided in Tables 2 and 3. As seen, all variables fall within their expected ranges, confirming the accuracy of the calculations.

## Analysis of empirical results

### Benchmark regression results

In the benchmark regression, we estimate equation (2) using the generalized method of moments (GMM) framework, specifically employing Difference GMM and System GMM. The results are shown in columns (1) through (4) of Table 4. The Difference GMM method relies on moment conditions for estimation, avoiding stringent assumptions about the distribution of variables and disturbances. By constructing 'internal' instruments, it effectively addresses endogeneity concerns related to reverse causality and omitted variables. System GMM extends Difference GMM by fully utilizing the horizontal equations, thereby improving the model's estimation efficiency. Accordingly, this study presents the System GMM estimation results in all regression tables, which form the basis for the analysis. Difference GMM results are also reported as a robustness check. Both methods are estimated using the two-step procedure. The AR(2) and Hansen test p-values exceed 0.1, confirming no second-order serial correlation in the residuals. This result validates all "internal" instrumental variables and satisfies the assumptions underlying both difference GMM and system GMM.

The estimation results in columns (1) to (4) of Table 4 show that, regardless of whether Difference GMM or System GMM is used for dynamic panel estimation, the coefficient for digital economy (Digital_pca) is significantly positive at the 1% level. This suggests that the digital economy effectively enhances agricultural TFP across Chinese provinces. Beyond statistical significance, the economic significance of the results is also substantial. Focusing on the estimates in column (4) of Table 4, the coefficient for Digital_pca is 0.12. This suggests that a one standard deviation (0.781) increase in digital economy development corresponds to a 0.094 increase in TFP_FE (0.781 * 0.012). This accounts for 18.54% of the variation in the sample (0.094/0.507, using the standard deviation of TFP_FE as the reference).

We also examine the estimated results for the control variables in column (4) of Table 4. The coefficient for urbanization (URBAN) is significantly positive, suggesting that as

**Table 2. Variable definitions.**

| Variables | Definition | Variable types | Unit |
|---|---|---|---|
| TFP_OLS | Agricultural TFP estimated using the Solow residual approach | Continuous | – |
| TFP_FE | Agricultural TFP estimated using the fixed-effects approach | Continuous | – |
| Digital_pca | The level of digital economy development, as estimated using principal component analysis | Continuous | – |
| Digital_szf | The digital economy development level, estimated using the entropy weight method | Continuous | – |
| URBAN | Urbanization | Continuous | %/100 |
| IU | Industrial upgrading | Continuous | – |
| OPEN | The degree of dependence on foreign trade | Continuous | %/100 |
| IA | The share of irrigated area | Continuous | %/100 |
| HUM | Rural human capital | Continuous | Year |
| DISA | The share of crop disaster-affected area | Continuous | %/100 |
| GOV | Government investment | Continuous | Ten thousand yuan per thousand hectares |
| INNO | Regional innovation capacity | Continuous | – |
| MKT | Level of marketization | Continuous | – |

Note: The data for the variables were calculated and compiled by the author.

**Table 3. Descriptive statistics for the variables.**

| Variables | N | Mean | Sd | Min | Max |
|---|---|---|---|---|---|
| TFP_OLS | 360 | 6.74e-11 | 0.338 | -2.503 | 2.383 |
| TFP_FE | 360 | 9.571 | 0.507 | 6.795 | 12.01 |
| Digital_pca | 360 | 7.52e-10 | 0.781 | -0.992 | 3.682 |
| Digital_szf | 360 | 0.140 | 0.118 | 0.013 | 0.752 |
| UI | 360 | 2.395 | 0.126 | 2.133 | 2.838 |
| OPEN | 360 | 0.265 | 0.287 | 0.002 | 1.548 |
| URBAN | 360 | 0.593 | 0.120 | 0.350 | 0.896 |
| IA | 360 | 0.528 | 0.328 | 0.179 | 3.117 |
| HUM | 360 | 7.820 | 0.635 | 5.923 | 9.915 |
| DISA | 360 | 0.177 | 0.161 | 0.006 | 1.464 |
| LGOV | 360 | 7.238 | 0.877 | 5.556 | 11.06 |
| INNO | 360 | 29.18 | 10.67 | 15.78 | 65.49 |
| MKT | 360 | 8.150 | 1.946 | 3.359 | 12.86 |

new-type urbanization progresses, agricultural TFP in China improves. The coefficient for industrial structure upgrading (UI) is significantly positive, indicating that industrial upgrading has fostered technological progress in Chinese agriculture. The estimated coefficient for trade dependence (OPEN) is significantly negative, aligning with the historical reliance of China's agricultural sector on foreign imports. This excessive dependence has constrained the potential of the domestic market, thereby suppressing agricultural productivity. The coefficient for the share of irrigated area (IA) is negative, indicating that the current level of irrigation in China may exceed the land's optimal capacity, thereby reducing agricultural TFP. The estimated coefficient for rural human capital (HUM) is negative, which appears counterintuitive. However, despite an increase in the average education level of rural labor during the study period, many workers rarely engage in agricultural activities. In fact, many retain

**Table 4. Benchmark regression results.**

| | (1)TFP_OLS | (2)TFP_FE | (3)TFP_OLS | (4)TFP_FE |
|---|---|---|---|---|
| | DIF-GMM | SYS-GMM | DIF-GMM | SYS-GMM |
| L.TFP_OLS | 0.489 | | -0.130*** | |
| | (1.17) | | (-2.72) | |
| L.TFP_FE | | -0.261*** | | 0.427*** |
| | | (-20.31) | | (49.00) |
| Digital_pca | 1.873* | 0.489*** | 0.665*** | 0.120*** |
| | (1.94) | (4.62) | (2.90) | (5.21) |
| URBAN | 3.070 | 17.723*** | 3.824*** | 0.382*** |
| | (0.53) | (18.10) | (5.93) | (3.21) |
| UI | -3.648 | 5.353*** | 2.036** | 2.626*** |
| | (-0.63) | (12.06) | (2.26) | (8.08) |
| OPEN | -0.598 | -0.0404 | -0.790** | -0.322*** |
| | (-0.59) | (-0.72) | (-2.51) | (-8.10) |
| IA | 0.102 | -0.572*** | -0.198 | -0.338*** |
| | (0.12) | (-10.27) | (-0.59) | (-5.24) |
| HUM | -0.901 | -0.893*** | -0.589*** | -0.340*** |
| | (-1.38) | (-10.96) | (-3.81) | (-13.07) |
| DISA | -0.222 | -0.153 | 0.0698 | 0.194** |
| | (-0.30) | (-1.04) | (0.37) | (2.29) |
| LGOV | -1.666* | 0.755*** | -0.553*** | -0.0271 |
| | (-1.69) | (14.85) | (-2.65) | (-0.54) |
| _cons | | | 3.332 | 2.467*** |
| | | | (1.37) | (5.43) |
| Province effect | Yes | Yes | Yes | Yes |
| Time effect | Yes | Yes | Yes | Yes |
| AR(2) | 0.969 | 0.748 | 0.682 | 0.105 |
| Hansen | 0.936 | 0.144 | 0.764 | 0.579 |
| N | 300 | 300 | 330 | 330 |

Note: ① *,**, and ***represent significance at the 10%, 5%, and 1% levels, respectively. The values in parentheses are t-statistics. ② AR(2) and Hansen tests report the corresponding p-values. The same applies below.

rural household registrations but work in urban areas, which likely explains the absence of a positive impact on agricultural TFP. The coefficient for the share of crop disaster area (DISA) is positive. A possible explanation is that a larger disaster-affected area may create an incentive for agricultural workers to enhance production techniques and increase yield per unit area, thereby mitigating the uncertainties of climate risks. The coefficient for government expenditure (LGOV) is not significant, indicating no substantial relationship between government investment and agricultural TFP.

## Robustness check

To further strengthen the robustness of our results, we substitute the composite method for the core explanatory variable with the entropy method, while keeping the sub-indicators unchanged. The resulting measure of digital economy development is labeled Digital_eem. The results in Table 5 demonstrate that, regardless of the estimation method, the coefficient for Digital_eem is significantly positive at the 5% level or better. This reinforces the robustness

**Table 5. Robustness check.**

| | (1)TFP_OLS | (2)TFP_FE | (3)TFP_OLS | (4)TFP_FE |
| | SYS-GMM | SYS-GMM | SYS-GMM | SYS-GMM |
|---|---|---|---|---|
| L.TFP_OLS | 0.230** | | 0.116*** | |
| | (2.30) | | (5.28) | |
| L.TFP_FE | | -0.401*** | | 0.280*** |
| | | (-25.89) | | (7.76) |
| Digital_eem | 0.749** | 4.518*** | 0.755** | 1.643*** |
| | (1.99) | (9.09) | (2.42) | (9.24) |
| URBAN | 0.596 | 29.434*** | 2.421*** | -1.201*** |
| | (0.68) | (10.70) | (3.91) | (-3.51) |
| UI | -4.569*** | 3.836*** | -5.152*** | -0.0917 |
| | (-3.81) | (5.07) | (-9.05) | (-0.18) |
| OPEN | 0.0292 | -0.0298 | 0.594*** | -0.343*** |
| | (0.27) | (-0.53) | (4.91) | (-6.89) |
| IA | -0.694** | -0.574*** | -0.647*** | -0.633*** |
| | (-2.44) | (-5.44) | (-7.21) | (-3.49) |
| HUM | 0.115** | -0.733*** | 0.152** | 0.0800*** |
| | (1.98) | (-10.42) | (2.47) | (2.97) |
| DISA | -0.607*** | 0.0480 | -0.331** | -0.936*** |
| | (-4.34) | (0.26) | (-1.97) | (-4.83) |
| LGOV | -0.0166 | 0.688 | 0.190*** | 0.219*** |
| | (-0.08) | (5.47) | (3.61) | (3.52) |
| _cons | | | 8.142*** | 6.397*** |
| | | | (5.89) | (6.78) |
| Province effect | Yes | Yes | Yes | Yes |
| Time effect | Yes | Yes | Yes | Yes |
| AR(2) | 0.303 | 0.967 | 0.564 | 0.738 |
| Hansen | 0.852 | 0.158 | 0.645 | 0.419 |
| N | 300 | 300 | 330 | 330 |

of the previous findings, confirming that the digital economy does indeed contribute to the enhancement of agricultural TFP in China.

## Mechanism check

The preceding tests confirm the positive impact of the digital economy on agricultural TFP in China. How, then, does the digital economy exert its positive impact? To answer this, we build on the earlier theoretical analysis and apply the approach of Beck et al. to identify the mechanism through which the digital economy affects agricultural TFP [36].

The estimation results in columns (1) and (2) of Table 6 focus on the coefficient for digital economy development (Digital_pca). This coefficient captures the net effect of the digital economy on regional innovation capacity (INNO) and marketization level (MKT), respectively. Drawing on the findings of Usman et al. and Villoria, both technological innovation and marketization positively influence agricultural TFP [37,38]. As shown in Table 6, the level of digital economy development (Digital_pca) significantly enhances both regional innovation capacity (INNO) and marketization level (MKT). These results support the view that the digital economy boosts agricultural TFP through its effects on innovation and marketization.

**Table 6. Mechanism check.**

|  | (1)INNO | (2)MKT |
|---|---|---|
|  | SYS-GMM | SYS-GMM |
| L.INNO | 0.638*** |  |
|  | (5.24) |  |
| L.MKT |  | 0.910*** |
|  |  | (28.07) |
| Digital_pca | 4.434*** | 1.006*** |
|  | (3.63) | (8.95) |
| URBAN | 17.159 | -1.545 |
|  | (1.52) | (-1.29) |
| UI | -24.03 | -0.986 |
|  | (-1.08) | (-0.79) |
| OPEN | -1.196 | -0.458*** |
|  | (-0.94) | (-3.39) |
| IA | 2.073 | -0.0605 |
|  | (0.97) | (-0.40) |
| HUM | 0.908 | -0.586*** |
|  | (1.01) | (-7.18) |
| DISA | 4.954 | -1.536*** |
|  | (1.43) | (-4.33) |
| LGOV | 1.122 | -0.619*** |
|  | (0.36) | (-5.45) |
| _cons | 43.21 | 13.95*** |
|  | (1.57) | (6.05) |
| Province effect | Yes | Yes |
| Time effect | Yes | Yes |
| AR(2) | 0.293 | 0.739 |
| Hansen | 0.139 | 0.278 |
| N | 330 | 330 |

## Heterogeneity analysis

**Regional heterogeneity.** The regional disparity in economic development across China has been a long-standing and well-documented phenomenon. To assess whether the impact of the digital economy on agricultural TFP varies by region, this study divides China's 30 provinces into eastern and central-western regions, following the classification criteria of the National Development and Reform Commission. The results of the subsample analysis, based on System GMM estimation, are presented in Table 7.

The regional analysis in columns (1) to (4) of Table 7 shows that the impact of the digital economy on agricultural TFP varies across different regions of China. Specifically, while the coefficient for Digital_pca is insignificant in columns (2) and (4), a comparison of columns (1) and (3) reveals that the coefficient is notably larger in the central-western regions. This suggests that the digital economy may have a stronger positive effect in these areas. The strength of the positive effect depends on several factors. While the eastern region enjoys higher levels of digital economy development, agricultural progress, and talent concentration, agriculture is not the primary focus of development in these provinces. Consequently, the digital economy has not significantly boosted agricultural TFP in the eastern region. This effect can be

**Table 7. Regional heterogeneity.**

| | (1)TFP_OLS | (2)TFP_FE | (3)TFP_OLS | (4)TFP_FE |
| --- | --- | --- | --- | --- |
| | SYS-GMM | SYS-GMM | SYS-GMM | SYS-GMM |
| | East | East | Central and Western regions | Central and Western regions |
| L.tfp_ols2 | 0.151 | | -0.263 | |
| | (0.53) | | (-1.54) | |
| L.tfp_fe2 | | -0.711 | | 0.222 |
| | | (-1.30) | | (0.55) |
| Digital_pca | 0.247** | 0.191 | 1.949** | 1.839 |
| | (2.22) | (0.22) | (2.20) | (0.70) |
| Urban | -0.00387 | -0.0185 | -0.0211 | 0.267** |
| | (-0.08) | (-0.10) | (-0.33) | (2.07) |
| UI | -4.042 | 1.969 | -6.685** | 0.663 |
| | (-0.54) | (0.09) | (-2.35) | (0.17) |
| OPEN | -0.171 | -0.0575 | -0.217 | -0.611 |
| | (-0.50) | (-0.07) | (-1.05) | (-1.34) |
| IA | -3.880 | 1.838 | -0.0788 | -0.974 |
| | (-1.00) | (0.19) | (-0.22) | (-1.29) |
| HUM | -0.299 | -1.080 | 0.582 | -0.0131 |
| | (-0.53) | (-0.56) | (0.75) | (-0.02) |
| DISA | -0.0696 | 1.486 | -2.574*** | -3.124*** |
| | (-0.08) | (0.19) | (-4.31) | (-3.04) |
| LGOV | 1.326 | -0.318 | 1.501*** | 1.275 |
| | (0.78) | (-0.06) | (2.84) | (1.40) |
| _cons | 5.409 | 23.29 | 4.388 | -10.81 |
| | (0.98) | (0.99) | (1.36) | (-1.32) |
| Province effect | Yes | Yes | Yes | Yes |
| Time effect | Yes | Yes | Yes | Yes |
| AR(2) | 0.933 | 0.496 | 0.253 | 0.595 |
| Hansen | 1.000 | 1.000 | 0.587 | 0.368 |
| N | 121 | 121 | 209 | 209 |

attributed to several factors. First, the central and western regions have long been crucial to China's agricultural production. For example, provinces in the western region, such as Guangdong, Guangxi, Sichuan, and Yunnan, are major producers of sugarcane, a key crop, while the central region is home to the country's most important grain-producing areas and livestock farming bases. Second, the digital economy in these regions has entered a "fast track." For instance, Guizhou province in the west established China's first big data pilot zone in 2015. This has not only allowed Guizhou to rapidly develop its digital economy but also enabled neighboring areas to benefit from this progress. The combined effect of these two factors leads to a stronger positive impact of the digital economy on agricultural TFP in the central and western regions.

To further investigate the marginal effect of the digital economy on agricultural TFP, this study divides the full sample into two subsamples—higher and lower agricultural TFP—based on the median TFP, and performs empirical tests for each subsample. The estimation results in Columns (1)–(4) of Table 8 closely align with those in Table 7. The coefficient of Digital_pca is insignificant in Column (2) but significantly negative in Columns (1), (3), and (4). Notably, the coefficient in Column (3) is larger in magnitude than in Column (1). These results suggest

**Table 8. Heterogeneity in productivity levels.**

| | (1)TFP_OLS | (2)TFP_FE | (3)TFP_OLS | (4)TFP_FE |
|---|---|---|---|---|
| | SYS-GMM | SYS-GMM | SYS-GMM | SYS-GMM |
| | Higher Agricultural TFP | Higher Agricultural TFP | Lower Agricultural TFP | Lower Agricultural TFP |
| L.tfp_ols2 | -0.185* | | -0.951*** | |
| | (-1.92) | | (-7.83) | |
| L.tfp_fe2 | | -0.170** | | -0.854*** |
| | | (-2.35) | | (-3.85) |
| Digital_pca | 0.316*** | 0.0289 | 0.749*** | 2.285*** |
| | (2.64) | (0.30) | (2.68) | (2.90) |
| Urban | -0.0315 | -0.00742 | -0.0809** | -0.122*** |
| | (-1.41) | (-1.21) | (-2.07) | (-3.15) |
| UI | 3.160** | 1.220 | -5.950** | 4.804*** |
| | (2.50) | (1.13) | (-2.42) | (2.63) |
| OPEN | -0.119 | -0.242*** | -0.544** | 0.0446 |
| | (-0.81) | (-2.65) | (-2.22) | (0.75) |
| IA | -2.088*** | -0.886** | -0.0580 | 0.856 |
| | (-3.09) | (-2.17) | (-0.10) | (1.63) |
| HUM | -1.018*** | -0.521*** | 0.00729 | 0.0322 |
| | (-10.36) | (-8.70) | (0.01) | (0.19) |
| DISA | -0.618** | -1.213*** | -4.871*** | -0.241 |
| | (-2.19) | (-3.00) | (-3.00) | (-0.63) |
| LGOV | -0.0661 | 0.585*** | 0.933* | 0.687*** |
| | (-0.26) | (3.38) | (1.89) | (3.12) |
| _cons | 4.486*** | 10.14*** | 12.79*** | 8.108** |
| | (3.37) | (5.62) | (4.47) | (2.48) |
| Province effect | Yes | Yes | Yes | Yes |
| Time effect | Yes | Yes | Yes | Yes |
| AR(2) | 0.671 | 0.327 | 0.385 | 0.136 |
| Hansen | 0.550 | 0.143 | 0.879 | 0.445 |
| N | 164 | 166 | 166 | 164 |

that the digital economy has a stronger positive impact in regions with lower agricultural TFP. This suggests diminishing marginal effects of the digital economy on agricultural TFP. The widespread adoption of digital technologies thus helps reduce regional disparities and fosters more balanced agricultural development across China.

## Discussion

### (1) The development of the digital economy enhances agricultural TFP

This paper advances research on digital technology and agricultural productivity, offering robust external validity relative to existing studies. The digital economy enhances agricultural TFP in both developing countries, such as China, and advanced economies. Deichmann et al. analyzed the impact of digital technology on agriculture in developing countries [39]. They concluded that digital technology promotes economic inclusivity and boosts agricultural productivity by complementing inputs such as labor and capital. Bocean found that in advanced economies within the European Union, agricultural digitalization strengthened resilience and sustainability while significantly increasing productivity [40]. The digital economy's impact

on agricultural TFP is universal, as global digitalization continues to deepen and expand its application in agriculture. Emerging technologies such as the Internet of Things and block-chain will drive the digital economy to further integrate the agricultural value chain, fostering smarter, more automated, and sustainable production. Agricultural producers in both developing and developed countries will increasingly depend on digital technologies to boost productivity, cut costs, and manage market risks. This will further strengthen the digital economy's universal impact on agricultural TFP.

### (2) The digital economy's positive impact on agricultural TFP is more pronounced in the central and western regions, as well as in areas with lower TFP

The heterogeneity findings of this study align with existing literature, showing that the digital economy's positive impact on agricultural TFP is stronger in the central and western regions, as well as in areas with lower TFP. Gong examined agricultural productivity trends across China's regions and found that most regions (23 of 28 provinces) and products (19 of 23) have not achieved convergence. Improvements in irrigation, education, and technology in lagging provinces could help narrow regional productivity gaps [41]. Deng et al. reached similar conclusions, highlighting that regions with less developed agriculture have an advantage in adopting new technologies and translating them into productivity gains [42]. These regions have significant potential to improve efficiency and related indicators through technological advances. In contrast, eastern regions face constraints, such as limited land resources, that hinder further marginal gains. This finding offers new empirical evidence for studies in developing countries like China, though it may not align with trends observed in some developed countries. Plastina et al. examined the impact of technological progress on agricultural TFP in the United States [43]. They found the strongest positive effect in the central region, followed by the western region, with the weakest effect in the east. This pattern reflects the high overall level of agricultural development and the more balanced spatial distribution in developed countries like the U.S. The central region, as the primary grain-producing area with commodity crop agriculture, benefits most from advanced technologies, while the eastern and western regions, which focus on dairy farming and specialty agriculture, experience a smaller impact.

### (3) The digital economy impacts agricultural TFP by boosting regional innovation and fostering marketization

The development of information technology broadens farmers' access to information, offering more learning and training opportunities that enhance their production skills. This finding is supported by studies such as Deichmann et al. [39]. Additionally, other scholars have shown that it promotes human capital accumulation, which in turn drives overall agricultural productivity [44]. The findings in the mechanism section — that the digital economy boosts agricultural TFP by enhancing regional innovation and promoting marketization—extend prior research with a more general conclusion. On one hand, the digital economy enhances data-driven precision in information matching and market responses, fostering innovative business models such as agricultural e-commerce platforms, crowdfunding for agricultural products, and the agricultural sharing economy. This, in turn, boosts agricultural TFP. This finding expands on the role of information technology in broadening farmers' access to information, reducing information asymmetry, and facilitating marketization. On the other hand, the digital economy fosters an innovative ecosystem that promotes collaboration among various stakeholders, optimizes the allocation of innovation resources, and generates knowledge spillovers. This supports regional economic transformation and sustainable development,

going beyond mere human capital accumulation to enhance regional progress across multiple dimensions. Thus, it offers a more comprehensive view of the digital economy's role in promoting human capital aggregation.

## Conclusion and policy implications

Against the backdrop of the flourishing digital economy, this paper conducts an empirical analysis to scrutinize the impact of the digital economy on overall factor productivity in agriculture. The research employs both the Difference GMM and System GMM methods, utilizing panel data from 30 provinces in China over the extended period from 2011 to 2022.

The findings of this study affirm that the digital economy significantly enhances the overall factor productivity level in China, and this conclusion withstands rigorous testing for robustness, affirming its stability and reliability under various analytical scenarios. The heterogeneity tests further reveal that the positive impact of the digital economy on agricultural overall factor productivity is more pronounced in the central and western regions, as well as in areas characterized by lower productivity. A detailed mechanism analysis suggests that the digital economy exerts a positive influence on agricultural overall factor productivity through the channels of innovative incentive effects and marketization effects. Based on the research findings, we propose the following policy recommendations:

Firstly, enhancing institutional and mechanistic reforms is imperative to cultivate an external environment conducive to comprehensive agricultural productivity through the facilitation of the digital economy. The government is urged to enact fiscal subsidies and tax relief policies to encourage participation from agricultural enterprises, academic institutions, industrial bases, and the agricultural populace, fostering engagement in the digitization of agricultural processes.

Secondly, diverse regions should meticulously chart the trajectory of agricultural development by considering their unique resource endowments and the state of advancement in the digital economy. Strategies should be tailored to align with local conditions. On one facet, regional authorities ought to judiciously allocate resources in accordance with their specific circumstances. This involves enhancing welfare provisions for digital professionals, and consolidating the foundational aspects of digital expertise in rural locales. Conversely, by synergistically integrating the digital economy with locally distinctive agricultural products, there is the potential to enhance the value-added attributes of these products.

Thirdly, it is imperative to enhance the sophistication of the digital dynamic mechanism for agricultural development, thereby unlocking the latent potential inherent in agricultural growth. The government is compelled not only to fortify the agricultural science and technology innovation system but also to improve essential infrastructure, encompassing elements such as roads, railways, internet connectivity, big data infrastructure, and cloud computing capabilities. Furthermore, optimizing the business environment, profoundly evolving market reforms related to agricultural factors, and consolidating the dividends bestowed by the digital economy upon agricultural development are also crucial.

## Acknowledgments

We would like to thank the reviews for providing professional comments on the manuscript

## Author contributions

**Conceptualization:** Jing He, Xiaokai Lei.

**Data curation:** Xiaokai Lei.

**Formal analysis:** Xiaokai Lei.

**Investigation:** Zhuangyu Wei.

**Methodology:** Zhuangyu Wei.

**Resources:** Zhuangyu Wei.

**Software:** Zhuangyu Wei.

**Supervision:** Jing He.

**Visualization:** Jing He.

**Writing – original draft:** Jing He.

**Writing – review & editing:** Jing He.

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
