## [Decision Letter · Decision Letter 0]

9 Oct 2024

PONE-D-24-30944Unveiling the Digital Revolution: Catalyzing Total Factor Productivity in AgriculturePLOS ONE

Dear Dr. Wei,

Thank you for submitting your manuscript to PLOS ONE. After careful consideration, we feel that it has merit but does not fully meet PLOS ONE’s publication criteria as it currently stands. Therefore, we invite you to submit a revised version of the manuscript that addresses the points raised during the review process.

**ACADEMIC EDITOR: **
**Experts in the field have reviewed your manuscript and you are expected to address their comments as early as possible. Thank you.**

We look forward to receiving your revised manuscript.

Kind regards,

Olutosin Ademola Otekunrin

Academic Editor

PLOS ONE

**Journal Requirements:**

3. Please amend the manuscript submission data (via Edit Submission) to include author Dr. Jun Wen.

Reviewers' comments:

Reviewer's Responses to Questions

Comments to the Author

1. Is the manuscript technically sound, and do the data support the conclusions?

Reviewer #1: Partly

Reviewer #2: Yes

2. Has the statistical analysis been performed appropriately and rigorously? 

Reviewer #1: No

Reviewer #2: Yes

3. Have the authors made all data underlying the findings in their manuscript fully available?

Reviewer #1: Yes

Reviewer #2: Yes

4. Is the manuscript presented in an intelligible fashion and written in standard English?

Reviewer #1: Yes

Reviewer #2: Yes

5. Review Comments to the Author

**Reviewer #1: ** Title: Unveiling the Digital Revolution: Catalyzing Total Factor Productivity in Agriculture

The paper tackles an important and relevant topic, examining the impact of the digital economy on agricultural Total Factor Productivity (TFP). While the literature review is fairly comprehensive and the methodology addresses endogeneity concerns through standard econometric techniques, there are several key issues that affect the contribution and strength of the study. Below are the main concerns that should be addressed:

•The measurement of the dependent variable, TFP, is crucial to the study, yet it lacks sufficient explanation. The authors use Stochastic Frontier Analysis (SFA) with only two inputs: the number of agricultural workers and fixed capital stock. This approach may miss critical inputs such as land, water usage, and energy, which are particularly important in agriculture. The paper should provide more justification for using SFA and discuss its strengths and limitations. It should also explain why alternative methods such as Data Envelopment Analysis (DEA) or Growth Accounting were not considered. This would strengthen the credibility of the TFP measure.

•The capital input is measured using the Perpetual Inventory Method (PIM), but the paper does not specify key details such as the depreciation rate used or how the initial value (base year) was chosen. The authors should provide clarity on these assumptions and perform a sensitivity analysis to ensure the robustness of the capital measurement.

•The paper measures the digital economy using broad indicators like internet penetration and e-commerce, which are comprehensive but may not fully capture the digital transformation in the agricultural sector. The authors should consider including agriculture-specific digital indicators, such as the adoption of digital farming technologies, rural digital infrastructure, and access to digital finance for farmers.

•The paper does not discuss potential biases that might arise when estimating the effect of the digital economy on TFP at the provincial level. It's unclear how the chosen control variables mitigate such biases. The authors should discuss potential biases in greater depth and explain how the chosen control variables (industrial structure, infrastructure, trade dependence, human capital, economic development, urbanization) help address these biases. More agriculture-specific control variables should be considered.

•While the control variables included are relevant and important for macroeconomic and structural factors, they may not be sufficient for isolating the impact of digital innovation on agricultural TFP. The current controls mainly focus on broader economic indicators rather than agriculture-specific ones. The paper should consider adding control variables that are more specific to agriculture, such as land quality, irrigation access, and farm size. These variables could provide a more accurate reflection of factors affecting TFP in agriculture.

•The paper includes a mechanism analysis but lacks agriculture-specific mechanisms. While regional innovation capability is a valuable addition, it would be beneficial to include other variables that directly measure how digital innovations impact agriculture. Consider adding variables like digital infrastructure in rural areas, agricultural R&D, farmer digital literacy, and use of digital finance to provide a more complete understanding of how digital innovations influence TFP.

•The study could benefit from robustness checks using alternative methods to measure TFP. This would enhance the validity of the results. The authors should consider using multiple methods for TFP measurement and compare the results to ensure robustness.

•The Hansen test yields a high p-value, but the paper does not discuss whether this could indicate overfitting. The authors should address whether the high p-value in the Hansen test suggests potential overfitting and, if so, how this issue was mitigated.

While the paper addresses an important topic, it is not yet ready for publication. The study's main weakness lies in the measurement of TFP and the lack of sufficient discussion on control variables, both of which are critical for ensuring the validity of the findings. I recommend a significant revision of the methodology, with particular attention to the measurement of TFP and the selection of variables, before reconsideration for publication.

**Reviewer #2: ** Comments

Thank you for allowing me to review this paper.

Overall, this is an interesting paper, addressing unveiling the Digital Revolution: Catalyzing Total Factor Productivity in Agriculture in China.

The article has a potential for publication, but in my view, the current version needs some revision to enhance its empirical contribution.

The paper must make an effort on the formatting. Please make a stronger effort in this regard.

No references were cited in the introduction section. Authors must provide the references.

Review the formatting in the literature review section line 3: services(Crawford , 2018); ligne 7: post-2012(Xu and Zhang, 2020) et line 10: context(Zhao et al., 2020); line 15 below(Bo and Zhang, 2021); line 22: enterprises(Chen and Hu, 2022); line 28: productivity(Gao, 2015); line 34: productivity(Liu et al., 2019); line 38: provinces(Li and Li, 2020); line 43: productivity landscape(Gong et al., 2020); line 53: agriculture(Wang et al., 2020); line 55: regions(Xue et al., 2020). Line 62: efficiency(Xia et al.,

2019).

Line 41 what does this mean productivity[10]? Is it a reference?

The formatting should be checked throughout the document.

Section hypothesis line 12: As China's agricultural development encounters bottlenecks, The simple infusion of factors alone is insufficient to meet the growing demand for high-quality agricultural products

Methodology

Suggestion: Authors should describe all variables used in regression analyses and their sources. They should also describe whether the variables used are ordinal, binary, or other.

The results are very interesting but lack a discussion with other work in the same field in China and elsewhere.

It will also be useful to specify the corresponding number (N) for each region in Table 6.

Reference:

6. PLOS authors have the option to publish the peer review history of their article (what does this mean? ). If published, this will include your full peer review and any attached files.

Do you want your identity to be public for this peer review? For information about this choice, including consent withdrawal, please see our Privacy Policy .

Reviewer #1: No

Reviewer #2: Yes: Bonna Antoinette TOKOU

---

## [Author Response · Author response to Decision Letter 0]

31 Dec 2024

1. The measurement of the dependent variable, TFP, is crucial to the study, yet it lacks sufficient explanation. The authors use Stochastic Frontier Analysis (SFA) with only two inputs: the number of agricultural workers and fixed capital stock. This approach may miss critical inputs such as land, water usage, and energy, which are particularly important in agriculture. The paper should provide more justification for using SFA and discuss its strengths and limitations. It should also explain why alternative methods such as Data Envelopment Analysis (DEA) or Growth Accounting were not considered. This would strengthen the credibility of the TFP measure.

Author's response: We sincerely thank the reviewer for highlighting the issues in our calculation of agricultural TFP. Upon reflection, we recognize that our initial approach lacked thorough consideration of agriculture-specific factors, such as land and agricultural machinery, which can significantly influence TFP. Additionally, our choice of methodology was not sufficiently comprehensive. Specifically, while the SFA method can estimate the rate of change in agricultural TFP, it does not directly provide the absolute level of TFP.

To address this issue, we made the following revisions: First, in calculating agricultural TFP, we drew on the methodologies of Lin(1992) and Gong(2018). Specifically, we selected input and output indicators as follows: the output indicator is defined as the total gross output value of agriculture, forestry, animal husbandry, and fisheries in each region (in million RMB). This value was adjusted to constant prices using the agriculture, forestry, animal husbandry, and fisheries GDP index with 2011 as the base year. The input indicators include total mechanical power (in 10,000 kW), fertilizer usage (in 10,000 tons), rural labor employment (in 10,000 people), and total sown crop area (in 1,000 hectares).

Second, regarding the calculation method, we note that both the DEA and SFA approaches can only provide the contemporaneous rate of change in TFP, rather than its absolute level. Since this study focuses on the impact of the digital economy on the level of agricultural total factor productivity (TFP), we adopted the classic Solow residual method and the fixed-effects method to measure agricultural TFP, denoted as TFP_OLS and TFP_FE, respectively. To minimize the influence of time-specific factors, we also controlled for time fixed effects during the estimation process.

Third, to enhance the robustness of the regression results, we report the estimation outcomes using both TFP_OLS and TFP_FE as dependent variables in all regressions.

Details of these revisions can be found in Section 4: Research design

References:

GONG, B. (2018), " Agricultural Reforms and Production in China: Changes in Provincial Production Function and Productivity in 1978-2015", Journal of Development Economics, Vol. 13218-31.

LIN, J. Y. (1992), "Rural Reforms and Agricultural Growth in China", American Economic Review, Vol. 82 No. 1, pp. 34-51.

2. The capital input is measured using the Perpetual Inventory Method (PIM), but the paper does not specify key details such as the depreciation rate used or how the initial value (base year) was chosen. The authors should provide clarity on these assumptions and perform a sensitivity analysis to ensure the robustness of the capital measurement.

Author's response: As noted in our response to Comment 1, we define the output indicator for agricultural TFP as the total gross output value of agriculture, forestry, animal husbandry, and fisheries in each region (in million RMB), adjusted to constant 2011 prices using the corresponding GDP index. For capital input, we replaced the perpetual inventory method with total mechanical power (in 10,000 kW) as a proxy for fixed capital investment in agricultural production.

3. The paper measures the digital economy using broad indicators like internet penetration and e-commerce, which are comprehensive but may not fully capture the digital transformation in the agricultural sector. The authors should consider including agriculture-specific digital indicators, such as the adoption of digital farming technologies, rural digital infrastructure, and access to digital finance for farmers.

Author's response: We greatly appreciate the reviewer’s insightful suggestion to incorporate agricultural digitalization metrics when evaluating the development of the digital economy. While this is an excellent idea, we attempted to incorporate it by collecting relevant data from China. Unfortunately, these indicators are either unsupported by available data or have limited disclosure periods. For instance, China’s Digital Rural Index has only been published since 2018. Using this data would significantly reduce the sample size for our study. After weighing the pros and cons, we believe our current indicator system represents the most optimal choice given the circumstances. We have also made every effort to evaluate the development of the digital economy comprehensively. To maximize the sample size, we extended the data coverage from 2011–2020 to 2011–2022, further enhancing the credibility and relevance of this study.

4. The paper does not discuss potential biases that might arise when estimating the effect of the digital economy on TFP at the provincial level. It's unclear how the chosen control variables mitigate such biases. The authors should discuss potential biases in greater depth and explain how the chosen control variables (industrial structure, infrastructure, trade dependence, human capital, economic development, urbanization) help address these biases. More agriculture-specific control variables should be considered.

Author's response:：In response to the reviewer’s comment, we made the following revisions: We added additional agriculture-related control variables to further mitigate potential omitted variable bias and enhance the uniqueness of the study. These adjustments also ensure that the model aligns more closely with the paper's core theme. The current set of control variables includes seven factors:

Urbanization (URBAN). The impact of urbanization on agricultural TFP can be seen in two ways. On one hand, it encourages labor migration from agriculture to non-agricultural sectors, driving improvements in production efficiency and resource utilization. Second, urbanization promotes technological diffusion and boosts market demand, thereby incentivizing agricultural innovation. On the other hand, urbanization may cause rural labor outflow and a reduction in arable land, leading to lower agricultural output and TFP. This study measures urbanization using the proportion of urban population to total regional population.

Industrial upgrading (IU). Industrial upgrading reallocates resources from the low-efficiency traditional agricultural sector to more productive modern industries. The growth of the secondary and tertiary sectors also offers increased technological support to agriculture, boosting agricultural TFP. We define industrial upgrading using the following equation (2).

(2)

The variable xi denotes the share of industry i’s output in total output.

The degree of dependence on foreign trade (OPEN). Import and export trade may influence agricultural TFP in both directions. A higher reliance on foreign trade allows China to benefit from technology spillovers, adopting advanced agricultural technologies and management practices from developed countries. Additionally, international market competition can drive efficiency gains and improve product quality in China’s agricultural sector. However, greater reliance on foreign trade may expose agricultural prices to international market fluctuations, undermining the stability of the sector. Import and export trade may also redirect resources to non-agricultural sectors, crowding out agricultural inputs. We measure foreign trade dependence as the ratio of total imports and exports to GDP, with trade values converted using the annual average exchange rate between the US dollar and the Chinese yuan.

The share of irrigated area (IA). An increase in the share of irrigated area improves water supply to farmland, enhances land use efficiency, and boosts crop yield and quality, thereby positively impacting agricultural TFP. However, excessive expansion of irrigated area may lead to water waste and other environmental issues, ultimately undermining agricultural TFP. We define IA as the ratio of irrigated farmland to total crop area

Rural human capital (HUM). Rural human capital impacts agricultural TFP in two key ways. First, high-quality human capital fosters the adoption and innovation of agricultural technologies, accelerates the spread of modern practices, and boosts productivity. Second, improvements in human capital can facilitate the shift from labor-intensive to knowledge-intensive agriculture, increasing the value added to production. Rural human capital is measured by the average years of education for the population aged 6 and older, as specified in equation (3).

(3)

In equation (3), pi represents the years of education for each level, and n denotes the number of education categories, set to 5. Specifically, we categorize the population's education level into five groups: illiterate or functionally illiterate（ ）, primary education（ ）, junior secondary education（ ）, senior secondary or vocational education（ ）, and tertiary education or higher（ ）.

The share of crop disaster-affected area (DISA). A higher proportion of crop disaster-affected area suggests significant damage to arable land, crops, and agricultural infrastructure. This, in turn, negatively affects agricultural output and resource allocation efficiency. We measure DISA as the ratio of affected crop area to total arable land.

Government investment (LGOV). Government investment in agriculture is primarily directed toward infrastructure development, technological research and development, and subsidies for agricultural products. By increasing such investment, production conditions improve, technological progress is accelerated, and agricultural TFP is enhanced. However, it is crucial to note that the effectiveness of government investment is shaped by factors such as policy design, implementation quality, and resource allocation. Over-reliance on direct subsidies or imprecise targeting can result in resource misallocation and incentive distortions in the agricultural sector, ultimately impeding improvements in agricultural TFP. We measure government investment as the expenditure on agriculture, forestry, and water affairs per unit of sown area. Given the high variability of this indicator, we apply a natural logarithm transformation.

As noted in the definition of the control variables, we included four agriculture-related variables and provided a detailed discussion of how each theoretically influences agricultural TFP. This serves as a direct and substantive response to the reviewer’s feedback. The specific revisions and their integration into the manuscript can be found in the main text.

5. While the control variables included are relevant and important for macroeconomic and structural factors, they may not be sufficient for isolating the impact of digital innovation on agricultural TFP. The current controls mainly focus on broader economic indicators rather than agriculture-specific ones. The paper should consider adding control variables that are more specific to agriculture, such as land quality, irrigation access, and farm size. These variables could provide a more accurate reflection of factors affecting TFP in agriculture.

Author's response: This issue is essentially the same as Point 4. The revisions we made in response can be found under “Author's Response” to Point 4.

6. The paper includes a mechanism analysis but lacks agriculture-specific mechanisms. While regional innovation capability is a valuable addition, it would be beneficial to include other variables that directly measure how digital innovations impact agriculture. Consider adding variables like digital infrastructure in rural areas, agricultural R&D, farmer digital literacy, and use of digital finance to provide a more complete understanding of how digital innovations influence TFP.

Author's response: Similar to Issue3, while the reviewer suggested incorporating agriculture-specific mechanisms, we have already included several agricultural variables as controls in the regression equation. More importantly, agriculture-related variables tied to the digital economy are unavailable, making it impossible to use them as mechanism variables. After extensive review of the literature and data, the current approach represents the best possible choice under the circumstances. It is worth noting that in the revised manuscript, we have also updated the data for the two mechanism variables—marketization and regional innovation capacity—through 2022. This further enhances the reliability of our findings.

7. The study could benefit from robustness checks using alternative methods to measure TFP. This would enhance the validity of the results. The authors should consider using multiple methods for TFP measurement and compare the results to ensure robustness.

Author's response: We have already addressed this issue in our response to Comment 1. In line with the reviewer’s suggestion, we report the regression results using both OLS and FE methods to estimate agricultural TFP as the dependent variable in all regression tables. This approach reduces the sensitivity of our findings to a single TFP measurement method, thereby enhancing robustness.

8. The Hansen test yields a high p-value, but the paper does not discuss whether this could indicate overfitting. The authors should address whether the high p-value in the Hansen test suggests potential overfitting and, if so, how this issue was mitigated.

While the paper addresses an important topic, it is not yet ready for publication. The study's main weakness lies in the measurement of TFP and the lack of sufficient discussion on control variables, both of which are critical for ensuring the validity of the findings. I recommend a significant revision of the methodology, with particular attention to the measurement of TFP and the selection of variables, before reconsideration for publication.

Author's response: We appreciate the reviewer’s comment on this issue. In the revised version of the manuscript, we have addressed this concern by adjusting the model specification to avoid a Hansen test p-value of 1 whenever possible. This improvement is reflected in most of our estimation results, with the only exception being Columns (1)–(2) in Table 7, where the Hansen test p-value remains 1. After reviewing the literature and conducting multiple tests on the Stata code, we identified the reason for this outcome. The issue can be summarized as follows: in the "Regional Heterogeneity" section, we divided the sample of China’s provinces into Eastern and Central/Western regions. After updating the sample period to 2022, the Eastern region had 121 observations, creating a disparity with the Central/Western regions. At the same time, we increased the number of control variables from six to seven in order to better examine the impact of the digital economy on agricultural TFP under constant conditions. However, this introduced a "flaw" in the estimation. Specifically, when using the SYS-GMM method, no matter how we adjust the model, the number of instrumental variables exceeds the number of cross-sectional observations in the Eastern region (11), which likely explains the Hansen p-value of 1. Although we could address this by reducing the number of control variables, we found that similar issues with Hansen/Sargan p-values of 1 or very close to 1 have appeared in a number of studies published in top journals（Dennis et al., 2008; Pathan and Faff, 2013）. After careful consideration, we believe that increasing the number of control variables, particularly agricultural controls, is more beneficial for improving the quality of our paper. The detail

---

## [Decision Letter · Decision Letter 1]

15 Jan 2025

Unveiling the digital revolution: Catalyzing total factor productivity in agriculture

PONE-D-24-30944R1

Dear Dr. Wei,

We’re pleased to inform you that your manuscript has been judged scientifically suitable for publication and will be formally accepted for publication once it meets all outstanding technical requirements.

Kind regards,

Olutosin Ademola Otekunrin

Academic Editor

PLOS ONE

Additional Editor Comments (optional):

Reviewers' comments:

Reviewer's Responses to Questions

**Comments to the Author**

1. If the authors have adequately addressed your comments raised in a previous round of review and you feel that this manuscript is now acceptable for publication, you may indicate that here to bypass the “Comments to the Author” section, enter your conflict of interest statement in the “Confidential to Editor” section, and submit your "Accept" recommendation.

Reviewer #1: All comments have been addressed

Reviewer #2: All comments have been addressed

2. Is the manuscript technically sound, and do the data support the conclusions?

Reviewer #1: (No Response)

Reviewer #2: Yes

3. Has the statistical analysis been performed appropriately and rigorously? 

Reviewer #1: (No Response)

Reviewer #2: Yes

4. Have the authors made all data underlying the findings in their manuscript fully available?

Reviewer #1: (No Response)

Reviewer #2: Yes

5. Is the manuscript presented in an intelligible fashion and written in standard English?

Reviewer #1: (No Response)

Reviewer #2: Yes

6. Review Comments to the Author

Reviewer #1: The authors have addressed all comments properly and made the necessary revisions to enhance the clarity and quality of the manuscript.

While the manuscript is well-revised, there is one suggestion that could further improve its readability and usefulness. Tables should be self-contained and provide all necessary information. It would be better to use full variable names instead of abbreviations in the tables presenting the model results.

Reviewer #2: (No Response)

7. PLOS authors have the option to publish the peer review history of their article (what does this mean? ). If published, this will include your full peer review and any attached files.

**Do you want your identity to be public for this peer review?** For information about this choice, including consent withdrawal, please see our Privacy Policy .

Reviewer #1: No

Reviewer #2: No

---

## [Editor Report · Acceptance letter]

PONE-D-24-30944R1

PLOS ONE

Dear Dr. Wei,

I'm pleased to inform you that your manuscript has been deemed suitable for publication in PLOS ONE. Congratulations! Your manuscript is now being handed over to our production team.

Kind regards,

on behalf of

Dr. Olutosin Ademola Otekunrin

Academic Editor

PLOS ONE